# The Role of Additive Manufacturing in Reducing Demand Volatility in Aerospace: A Conceptual Framework

Ageel Abdulaziz Alogla * , Ateyah Alzahrani and Ahmad Alghamdi

Industrial Engineering Department, College of Engineering in Al-Qunfudhah, Umm Al-Qura University, Mecca 21961, Saudi Arabia
* Correspondence: aaogla@uqu.edu.sa

**Abstract:** The aerospace industry faces challenges in managing inventory effectively due to long product life cycles and unpredictable demand. Additive Manufacturing (AM) is a promising technology that enables the on-demand production of spare parts, potentially reducing inventory costs and improving supply chain efficiency. This paper proposes a novel conceptual framework for employing AM in the aerospace spare parts industry to isolate demand volatility. A conceptual approach is employed in this study, which involves a comprehensive literature review to identify the factors to consider when employing AM for spare parts and the methods for demand volatility isolation, followed by a structured framework development that outlines the decision-making steps for AM utilization based on the identified factors. The framework outlines a structured approach for using AM to produce spare parts and isolate demand volatility, which can help mitigate the impact of demand uncertainty on inventory management. The proposed approach provides a basis for future research and has the potential to transform how spare parts are produced and managed in the aerospace industry. Overall, this paper contributes to the emerging literature on AM in the aerospace industry by presenting a novel approach to improving inventory management and addressing demand uncertainty.

**Keywords:** additive manufacturing; powder bed fusion; demand volatility; spare parts; aerospace





## 1. Introduction

Demand volatility refers to the degree of unpredictability or variability in customer demand for a product or service over a certain period of time [1,2]. It is a critical factor in supply chain management because it directly affects a company's ability to meet customer demand while minimizing inventory costs and avoiding stockouts [3]. In general, high demand volatility means that customer demand is unpredictable and subject to large fluctuations, while low demand volatility means that demand is more stable and predictable [4]. Managing demand volatility in the supply chain can be challenging because it requires balancing competing priorities such as minimizing inventory costs, reducing lead times, and improving customer service [5]. Strategies for managing demand volatility may include using demand forecasting tools [6], implementing flexible manufacturing processes [7], improving supplier collaboration [8], and optimizing inventory levels [9].

Demand volatility in the spare parts industry can occur due to several factors. One such factor is changes in product design [10]; hence, products go through different stages of their lifecycle and the demand for spare parts can fluctuate. For instance, as a product becomes obsolete or reaches the end of its lifecycle, the demand for spare parts may decline rapidly. Volatility in demand also occurs due to equipment failures [11], as the demand for spare parts can be unpredictable and can fluctuate depending on how often the equipment fails and how quickly the parts can be sourced and delivered. Shifts in customer demand as in the case of emerging trends for certain brands, and changes in market conditions such as economic downturns, changes in regulations, or changes in technology can also cause fluctuations in the demand for spare parts [12,13].

The aerospace spare parts industry in particular is subject to demand volatility due to several factors. The first of these is changes in aircraft design [14] that emerges from the introduction of new models or technologies, which may require different parts, resulting in changes in demand. The second factor is aircraft maintenance requirements, which cause demand volatility due to the unpredictable nature of service requests [15], which can lead to an uneven distribution of maintenance tasks over time [16], resulting in intermittency in spare parts demand. This intermittency can complicate spare parts inventory control [16], and lead to an increase in inventory costs and a backlog of goods [17]. This is because accurate prediction of the demand for spare parts can produce great economic and military returns [18]. The third factor is accidents or incidents that cause demand volatility in the aerospace spare parts industry. This is due to the complexity of modern equipment and the potential for serious consequences in the event of a failure [19]. The aviation industry is now in demand for properties that can eliminate fatal results of fire in case of aircraft accidents [20,21]. Aircraft maintenance-related accidents and serious incidents can also cause demand volatility [22].

The last critical factor that causes demand volatility in aerospace spare parts is changes in economic conditions. For instance, the outbreak of the pandemic has disrupted aerospace operations and consumer demand, leading to cash flow constraints for individual corporations [23]. This has necessitated the use of models that are appropriate for conditions of unstable demand, such as the calculation of safety stock to prevent shortage [24]. Additionally, external economic pressures and the attraction of outside investment capital have led to the increasing professionalization of Fixed Base Operators (FBOs) and the growth of larger corporate FBO chains operating in the market [25].

Aerospace spare parts demand volatility can be managed through a variety of strategies. An extended (T, s, S) inventory control strategy [26] can be used to determine supply time, consumption, and demand. Additionally, forecasting techniques such as the moving average, exponential smoothing, naive theory, support vector regression, feed-forward neural networks, and adaptive model selection methodologies [27] can be used to predict demand. A double-layer dynamic programming maintenance model [28] can be employed to obtain the optimal spare parts inventory and the optimal maintenance strategy. Finally, an integrated forecasting methodology based on failure and condition information [29] can be utilized to manage demand volatility.

However, the stochastic characteristics of the aircraft spare parts make it difficult to precisely predict future demand [30]. As a result, the inability to decide the proper inventory in accordance with the demand for equipment can lead to improper inventory structure, the inability of limited capital to produce maximum returns, and weaker equipment support ability [18]. Additionally, matching supply with demand in the aerospace spare parts industry is challenging due to the long life cycles of the primary products [31], the complexity of managing the many types of spare parts [32], the intermittent demand [33], and the stochastic nature of part's demand. The quality of the demand input parameters used for the calculation is also a key factor in determining the optimal allocation and consumption of spare parts [34].

Furthermore, there are other aspects of the issue of volatility in the spare parts demand that must be raised, beyond just the difficulty of matching supply with demand in the aerospace industry. Managing inventory in the aerospace spare parts industry can also be challenging due to the high value of the parts and the long lead times associated with manufacturing and delivery. Demand volatility can make inventory management even more complex, as companies must balance the cost of carrying inventory against the risk of stockouts. Spare parts stockout events caused by uncertainty can lead to severe losses [35]. Maintenance costs, especially spare parts costs, are also a concern [36]. Furthermore, uncertainty and unpredictability in spare parts demand can make traditional inventory control policies ineffective [37]. Finally, the lack of spare parts availability can cause equipment downtime, resulting in customer dissatisfaction and possible penalty costs for after-sales service providers [38].

The aerospace industry is highly dependent on the availability of spare parts to maintain the safety and reliability of aircraft. However, demand volatility in the industry can create significant challenges in managing inventory and supply chain risk. Additive Manufacturing (AM), also known as 3D printing, has emerged as a promising technology that can potentially mitigate the severe effects of demand volatility. Furthermore, Powder Bed Fusion, PBF, has emerged as a particularly powerful tool in aerospace manufacturing due to its high precision and the ability to fabricate complex geometries. For instance, in a recent review of metal AM in the aerospace industry, it was noted that the technology can offer significant advantages in terms of weight reduction, lead time reduction, and supply chain optimization [39,40]. However, the effective integration of AM into the spare parts industry requires a structured approach that can isolate demand volatility and help mitigate the impact of demand uncertainty on inventory management. In this context, this study proposes a novel conceptual framework for employing AM in the aerospace spare parts industry to address demand uncertainty. This paper's main contribution is the development of a structured approach for using AM to produce spare parts and isolate demand volatility, which has the potential to transform how spare parts are produced and managed in the aerospace industry. In particular, this paper explores potential viable rollout scenarios in which AM can be employed in combination with Conventional Manufacturing (CM) methods to mitigate the effect of demand volatility and uncertainty.

The initial focus of this conceptual paper is to examine the economic reasons behind the adoption of AM, specifically regarding inventory costs and suggesting rollout scenarios that incorporate AM alongside traditional manufacturing processes in aerospace, and to outline some of the advantages related to supply chain implementations. The rest of the paper is organized as follows: Section 2 offers a detailed background on the challenge of optimizing spare parts inventory costs in relation to demand volatility. It also discusses the utility of AM in this context based on certain characteristics of this technology. Section 3 introduces two novel AM rollout scenarios in the aerospace spare parts industry. Section 4 finally discusses the introduced rollout scenarios and concludes the work by offering recommendations for future directions in this domain.

## 2. Demand Volatility in Spare Parts

### 2.1. Production and Inventory Planning

Inventory costs are the costs associated with storing goods in a warehouse, such as purchase fees, order fees, and storage fees. Other costs associated with inventory include fixed inventory holding costs and stockout costs [41], as well as production costs with fixed costs. Tradeoffs between these costs occur because minimizing one type of cost may increase another type of cost [42].

Ordering costs are the costs associated with placing and receiving an order, such as the cost of paperwork, processing, and transportation [43]. These costs can be reduced by ordering in larger quantities, but this may increase holding costs and the risk of obsolescence. Holding costs are the costs of storing and managing inventory, such as storage, handling, insurance, and obsolescence costs [44]. These costs can be reduced by reducing inventory levels, but this may increase ordering costs and the risk of stockouts. Shortage or stockout costs are the costs associated with not having enough inventory to meet customer demand, such as lost sales, backorders, or customer dissatisfaction [45]. These costs can be reduced by increasing inventory levels, but this may increase holding costs and the risk of obsolescence.

Tradeoffs between these costs occur because minimizing one type of cost may increase another type of cost. For example, ordering in larger quantities may reduce ordering costs but increase holding costs and the risk of obsolescence. Similarly, reducing inventory levels may reduce holding costs but increase stockout costs and the risk of stockouts. The optimal inventory level is determined by finding the balance between these costs. Techniques such as Economic Order Quantity (EOQ) or Economic Production Quantity (EPQ) can be used to find the optimal inventory level that minimizes total inventory costs by balancing the tradeoffs between ordering and holding costs [46] but neglects stockout costs, or assumes that it is not permitted. This

is because both EOQ and EPQ assume a deterministic demand nature; thus, stockout events should be prevented. If demand is characterized by volatility, then the problem becomes the newsvendor's problem. The interested reader is referred to the work of [47].

Economies of scale refer to the cost advantages that result from increasing the scale of production [48]. As the production quantity increases, the average cost per unit decreases due to factors such as specialization, automation, and increased purchasing power. However, the "buy-to-fly ratio" can also impact the overall cost-effectiveness of production. The buy-to-fly ratio represents the ratio of the weight of raw material purchased to the weight of the final product produced [49]. AM can help reduce the buy-to-fly ratio by enabling the production of complex parts with minimal waste. EOQ and EPQ calculations can help businesses identify the optimal production or order quantity that maximizes economies of scale while minimizing inventory costs and reducing the buy-to-fly ratio. By producing or ordering larger quantities, businesses can reduce the unit cost of production or ordering, leading to cost savings and improved profitability [50].

*2.2. AM Characteristics and Inventory Costs*

AM has several characteristics that can help in managing inventory costs [51]. One of the most significant benefits of AM is its ability to produce items on-demand, which can help to reduce the need for holding excess inventory [52]. Conventional Manufacturing (CM) methods require items to be produced in large batches, which can result in excess inventory that must be stored, managed, and accounted for. In contrast, AM allows for items to be produced as needed, reducing the need for holding inventory. Although there are no costs that can be spread out over larger production runs, AM still demonstrates economies of scale due to increased machine throughput [53] and operator learning [54].

Another relevant characteristic of AM is its design flexibility [55]. AM can produce items with complex geometries and structures that may not be possible with traditional manufacturing methods [56]. This can help to reduce the need for holding excess inventory to account for design variations. For example, a company could produce a range of items with different designs or sizes, without the need for holding excess inventory.

Rapid prototyping is another characteristic of AM [57]. Prototypes can be produced quickly and at a lower cost than traditional manufacturing methods [57], reducing the need for holding excess inventory to account for design changes. AM can also help to reduce tooling costs, as items can be produced without the need for expensive tooling or moulds [58]. This can make smaller production runs more economically feasible, reducing the need for holding excess inventory. Overall, AM has several characteristics that can help to reduce inventory costs. These include on-demand production, design flexibility, rapid prototyping, and reduced tooling costs. While AM may not be suitable for all applications, it can be a useful tool in managing inventory costs for markets characterized that need to produce items with high variability or low demand.

Examining various production contexts is crucial in evaluating the costs and benefits of using AM from a supply chain perspective. AM adoption has gone through different stages and can be classified into four distinct categories based on the production settings. The initial category involves using AM for toolmaking, where polymer moulds or metal dies are rapidly created using AM [59]. The second category uses AM in the product development phase, specifically for producing prototypes [60]. The third category involves using AM in products requiring mass customization or complex geometries [61,62], marking a shift toward the direct manufacturing of end-use products. The fourth and final category involves utilizing AM in production settings to improve the response to market demand volatility [63,64], representing a significant change in the motivation for AM adoption. Although AM's ability to produce customized parts is still a critical feature, its recent adoption in industries with higher production volumes is evident, especially in the spare parts industry [65,66], where AM is used to produce products with varying production volumes in response to unexpected disruptive customer orders. This work focuses on the fourth category and elucidates how AM

can be deployed in the aerospace spare parts industry to isolate demand variability effectively. The next section presents a novel framework for such a case.

### 2.3. AM in Aerospace Spare Parts

The role of AM technology has gained global attention due to its rapid growth and impact on spare parts inventory management. Despite significant benefits, many authors suggest that further investigation is necessary to improve the deployment of AM in aerospace and flight manufacturing. Mecheter et al. [67] provided a detailed review of the application of AM technology in the supply chain of spare parts, highlighting its significant potential benefits.

There is considerable belief among manufacturing experts that AM could be a game changer in the future of aerospace manufacturing [68]. This is due to the efficient use of raw materials in manufacturing spare parts. One of the popular methods in manufacturing is the PBF. In traditional milling processes, the raw material is often plates, which can result in a significant amount of recyclable waste, sometimes up to 95%. However, PBF produces parts with near-final contours, resulting in only around 5% waste. Even this waste can be reused after undergoing a screening and sorting process [69].

Mohanavel et al. [70] argued that the application of AM in aerospace industries can make robust and lightweight products in a competitive time compared to the conventional way of manufacturing. In addition, those authors mentioned that using AM for aerospace manufacturing is friendlier to the environment due to several pros, such as preventing the waste of raw materials and the overall safety of both humans and nature. Mohanavel, Ali, Ranganathan, Jeffrey, Ravikumar, and Rajkumar [70] discussed that using AM can result in strong and lightweight parts required for less fuel consumption for aerospace applications and automobile industries.

Reddy et al. [71] conducted a study to analyze the effects of implementing AM on the Flap Lever, a specific part of the aerospace industry. The study found that using AM provided several advantages over traditional manufacturing techniques, including weight optimization, cost savings in the buy-to-fly ratio, and reduced fuel costs and time to market. Moreover, the product was modelled using CATIA software, and the researchers discovered that AM made it easier to customize the product.

Another research study, conducted by Khajavi et al., as described in [65], aimed to investigate the impact of AM on the supply chain configuration in the aeronautics industry. Specifically, the authors focused on the spare parts supply chain of the F-18 Super Hornet fighter jet, as the air-cooling ducts of the environmental control system are manufactured using AM. The study analyzed multiple scenarios, including a comparison between the current centralized supplier practices and a decentralized production approach facilitated by AM technology. The researchers found that using AM technology under centralized production was the most favourable scenario for spare parts production. Furthermore, the study assessed the total operating cost as the parameter to compare between different scenarios. This research highlights the potential benefits of using AM for designing and producing spare parts in the aerospace industry, particularly in terms of optimizing the supply chain and reducing costs.

In another research study, conducted by Sgarbossa et al. [72], the impact of AM on spare parts inventory management was investigated, with a focus on identifying the advantages of AM compared to CM techniques. To analyze the impact of AM on spare parts inventory management, the author developed a decision tree approach that can aid decision-makers in selecting the optimal process to handle intermittent demands, long procurement lead times, and high downtime costs when the necessary spare parts are not available in a timely manner. The authors listed multiple outcomes and concluded that AM can generally improve profitability for small parts, low backorder costs, high procurement lead times of the counterpart produced via CM, and high review periods. This research emphasizes the potential benefits of utilizing AM in the spare parts inventory management

of aerospace manufacturing, and the decision tree approach can help practitioners make evidence-based decisions when considering AM for their inventory management needs.

Another study by Angela et al. [73] developed a framework that helps to assess the sustainability of AM in spare parts industries from a life cycle approach. The author mentioned that the proposed framework will help to have an overall impact on the sustainability of AM in spare part industries. In a similar vein, Cestana et al. [74] argued that the use of AM technology allows for the immediate printing of parts, which reduces the need for physical storage space and decreases inventory holding costs throughout the supply chain.

Lolli et al. [75] proposed a new inventory management model that considers a spare part installed on a pool of systems that can be managed with CM or AM. They show that printing on demand with a procured printer is not preferable if the inventory capacity is not constrained, due to the high production cost and the high acquisition cost of the 3D printer. Considering the actual cost, AM is preferable only for an outsourced make-to-stock system. Abidar and Chaabane [33] investigated the advantage of AM integration for spare parts optimization in a multi-echelon inventory system using three different scenarios. The first scenario considers the conventional case where there is no AM integration, and the second scenario considers AM integration only in the central maintenance centre (CMC). The third scenario assumes AM integration in CMC and Regional Maintenance Centers (RMCs). The result showed that the best scenario for AM manufacturing integration is to consider a decentralized AM location. Another study by Lastra et al. [76] investigated that AM capability can play an important role in industrial maintenance, especially in predictive maintenance. The result showed that AM plays an important role in predictive maintenance and will become more important in the future.

Knofius et al. [77] investigated the profitability of switching from CM to AM for low-volume spare parts businesses. The study showed that despite the higher unit price of AM, switching to AM yields an average cost saving of about 35%. Bacciaglia et al. [78] proposed a methodology for processing large 3D models to be applied before the slicing process. For this method, a case study was conducted with fixed UAV wings. The result showed that the production time can be reduced by 50%, which is very attractive for AM. The study argues that this will have an impact on the AM industry and lead to an increase in production volume and a reduction in the time required for the manufacturing process. Halvorsen and Lamvik [79] developed semi-structured interviews to assess knowledge needs for the adoption of AM for spare parts in the maritime industry. The authors found that AM can be applied in practice for spare parts in the maritime industry, but the adoption of AM is still at an early stage. Another research study, by Ghadge et al. [80], based on the data from the supply chain network of leading global airlines showed that the aircraft supply chain was significantly improved by implementing AM. In addition, the authors argued that AM helps to balance inventory levels and increase responsiveness while reducing disruptions and carbon emissions in supply networks.

However, despite the benefits of using AM in spare parts manufacturing, there are limitations based on Bacciaglia, Ceruti, and Liverani [78], who reviewed the previous application of large spare parts using AM. The author argued that AM has limitations with producing large-scale components in a single piece, limiting the number of connections, joints, and glueing. While prior work on AM in the aerospace industry has been valuable, it has not yet fully explored the viability of AM considering demand aspects such as volatility, which this paper addresses. Additionally, a research approach was implemented by Zhang et al. [81], who developed a simulation to analyze the impact of using AM to replace traditional inventory for an on-demand spare parts supply system. The simulation result showed that the use of AM is not cost-effective compared to traditional spare parts supply in the warehouse. In addition, some critical factors may affect the overall performance of AM operations. Table 1 presents the parameters of previous studies on AM and its impact on spare parts.

**Table 1.** List of studies that discussed the impact of AM on spare parts.

| Study | Method | | Result | | | AM Impact | |
|---|---|---|---|---|---|---|---|
| | Case Study | Review | Strong Suggestion | Suggest with Limitations | No Suggestion | Positive | Negative |
| Mecheter, Pokharel, and Tarlochan [67] | | ✓ | ✓ | ✓ | | ✓ | ✓ |
| Najmon, Raeisi, and Tovar [68] | ✓ | | ✓ | | | ✓ | |
| Mohanavel, Ali, Ranganathan, Jeffrey, Ravikumar, and Rajkumar [70] | ✓ | | ✓ | | | ✓ | |
| Reddy, Mirzana, and Reddy [71] | ✓ | | ✓ | | None | ✓ | |
| Khajavi, Partanen, and Holmström [65] | ✓ | | ✓ | | | ✓ | |
| Sgarbossa, Peron, Lolli, and Balugani [72] | ✓ | | ✓ | ✓ | | ✓ | ✓ |
| Cestana, Pastore, Alfieri, and Matta [74] | ✓ | | ✓ | | | ✓ | |
| Bacciaglia, Ceruti, and Liverani [78] | ✓ | | ✓ | ✓ | | ✓ | ✓ |
| Lolli, Coruzzolo, Peron, and Sgarbossa [75] | ✓ | | | ✓ | | ✓ | |
| Abidar and Chaabane [33] | ✓ | | ✓ | | | ✓ | |
| Lastra, Pereira, Díaz-Cacho, Acevedo, and Collazo [76] | ✓ | | ✓ | | | ✓ | |
| Knofius, van der Heijden, and Zijm [77] | ✓ | | ✓ | | | ✓ | |
| Bacciaglia, Ceruti, and Liverani [78] | ✓ | | ✓ | | | ✓ | |
| Halvorsen and Lamvik [79] | ✓ | | ✓ | | | ✓ | |
| Ghadge, Karantoni, Chaudhuri, and Srinivasan [80] | ✓ | | ✓ | | | ✓ | |
| Zhang, Jedeck, Yang, and Bai [81] | ✓ | | | | ✓ | | ✓ |

## 3. Framework for Aerospace Spare Parts Demand Volatility Isolation through AM

In this section, a framework is proposed for the decision-making process of employing AM in the aerospace spare parts industry to isolate volatility in demand. Figure 1 depicts the framework, which outlines the crucial factors that should be considered to determine which spare parts can benefit from AM to isolate volatility in demand.

First, three decisive factors should be considered to elucidate for what spare parts, if any, AM can be employed. The first factor, symbolized in the first Decide module, deals with the suitability of using AM for the surveyed parts. While AM is a relatively mature technology, it still lacks some capabilities that make certain products difficult to fabricate using AM [82]. If a product is difficult to fabricate using AM, then understandably the product must be made using CM methods. The second factor, represented by the second Decide module, concerns the level of demand fluctuation experienced by each product, and Olhager's [83] method can be utilized here to estimate the Coefficient of Variation (CV), which is a measure of demand volatility. The third decisive factor deals with the expected production volume, measured by the historical Annual Production Quantities (APQ).

Once the CV is over the threshold of 40%, AM can then be considered as a viable option, with distinct rollout scenarios, however. The third decisive factor would then help in choosing the right rollout scenario. If APQ is less than the Low Production Volume (LPV), then a combinational deployment strategy where AM is utilized for products with high volatility and where CM methods are employed to cater for products with stable demand is an effective option [84,85]. If the APQ, however, is more than the LPV, then AM

can be combined with CM methods for the same product. In this case, when the current production and inventory policy (e.g., continuous review policy) underestimates future demand volume, instead of incurring stockout penalties, AM can serve in this case [45] by building on the characteristic of on-demand production.

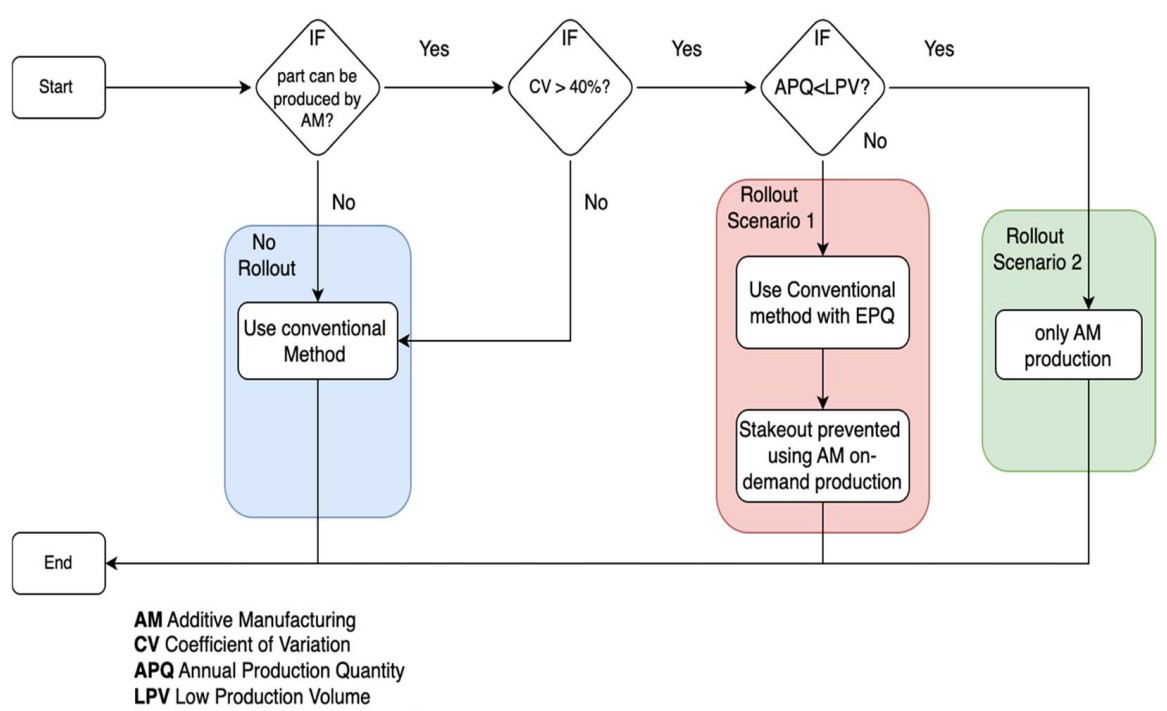

**Figure 1.** A framework to differentiate the distinct AM rollout scenarios in relation to demand volatility level and production volume.

The framework can be applied to various scenarios in the aerospace industry. For example, in the case of spare parts for aircraft engines, the framework can help identify which parts are suitable for AM fabrication and have high demand volatility. These parts can be produced using AM to reduce the lead time and minimize inventory costs. Another application of the framework is for spare parts used in the interior of aircraft, such as seating components or cabin panels. These parts may have lower annual production quantities but high demand variability due to changes in airline interior design or repair needs. The framework can help identify which parts can be produced using AM to reduce lead time and avoid stockout penalties. The framework can also be applied to the production of jigs and fixtures used in the manufacturing process of aerospace components. These jigs and fixtures may have low annual production quantities but high demand volatility due to changes in production processes or new product designs. AM can be used to produce these jigs and fixtures on demand, reducing the lead time and inventory costs.

A typical illustration of a spare part in the aerospace industry that has high demand volatility and is produced in low volumes is the turbine blade for aircraft engines. The turbine blades are critical components in the engine that experience high temperatures and stresses during operation and need to be replaced periodically due to wear and tear. However, the demand for turbine blades can be unpredictable and varies depending on factors such as flight schedules, maintenance schedules, and unexpected engine failures. Assume that a particular aircraft engine model has 1000 turbine blades installed in its fleet, and the expected lifespan of each blade is 10,000 flight hours. If the average flight time for

the engine is 5 h per day, then the expected lifespan of each blade would be approximately 5 years. However, due to the unpredictable nature of the demand for turbine blades, the actual number of blades required in any given year can vary significantly. For instance, one year, there may be a high number of engine failures that require turbine blade replacements, resulting in a demand of 300 blades. In another year, there may be no failures, resulting in a demand of only 50 blades.

In such a scenario, using the proposed framework can help identify which turbine blades can be produced using AM to reduce the lead time and minimize inventory costs. The framework first considers factors such as the complexity and size of the blade, and the material properties required to determine whether certain blades can be produced by AM (i.e., 1st decisive factor in Figure 1). If these requirements can be met with AM, and the CV is over the threshold of 40%, AM can be considered a viable option whether as a combinational method with forging or casting or as the only production method. In the aerospace industry, the production volume for turbine blades can be relatively low compared to other industries, due to the high level of customization required for each blade to meet specific performance requirements. It is not uncommon for turbine blades to be produced in batches of 100–500 units per year. In the case of historical production volumes showing a number less than 100, then AM can be employed as the only production; otherwise, it must be combined with forging or casting to prevent stockout events from occurring.

After describing the proposed framework for integrating AM into the aerospace spare parts industry, it is important to note that the certification process for additively manufactured aircraft parts is a critical aspect to ensure their safety and reliability. The certification process for additively manufactured aircraft parts involves a rigorous evaluation of the part's design, manufacturing process, and performance characteristics to ensure that it meets the airworthiness standards set by aviation regulatory bodies such as the Federal Aviation Administration (FAA) and the European Aviation Safety Agency (EASA) [86]. Part 21 of the EASA regulations sets out the requirements for the certification of aircraft parts, including additively manufactured parts. The Part 21 Product Organization Approval (POA) procedure is a certification process for organizations that design and manufacture aircraft parts, including additively manufactured parts [87]. The POA involves an evaluation of the organization's quality management system, design and manufacturing processes, and technical capabilities, among other factors. The certification process ensures that the parts meet the airworthiness standards set by EASA, which are designed to ensure the safety of the aircraft and its passengers. Thus, obtaining a Part 21 POA is a critical step in the certification process for additively manufactured aircraft parts.

While the proposed framework provides a systematic approach to decision-making regarding the adoption of AM in the aerospace spare parts industry to reduce demand volatility, several limitations need to be addressed. First, the framework assumes that the suitability of AM for the surveyed parts is known a priori. However, this may not always be the case, especially for new parts where the manufacturing process has not been established yet. Additionally, the framework does not consider the cost–benefit analysis of using AM for each spare part, which is a critical factor for decision-making. Moreover, the framework does not consider the impact of using AM on the supply chain, including the logistics and inventory management, which can significantly affect the cost and lead time of producing spare parts. Finally, the framework does not account for the environmental impact of using AM, which can be significant considering the amount of energy and resources required to produce spare parts using AM.

To address these limitations, future directions can be made in several areas. First, the suitability of AM for the surveyed parts can be assessed using machine learning algorithms that can predict the feasibility and cost-effectiveness of using AM for different parts based on their design specifications. Secondly, a comprehensive cost–benefit analysis can be conducted for each spare part to determine the economic viability of using AM. Thirdly, the impact of using AM on the supply chain can be analyzed using simulation tools that can model the logistics and inventory management processes. Fourthly, the environmental

impact of using AM can be quantified using life cycle assessment methods that consider energy and resource consumption, waste generation, and emissions associated with AM. Finally, the framework can be extended to include other factors that can affect the decision-making process, such as intellectual property rights, the level of customization required, and the lead time requirements. By addressing these limitations and incorporating future directions, the proposed framework can be further enhanced and applied to practical situations in the aerospace spare parts industry.

## 4. Conclusions

In conclusion, demand volatility refers to the unpredictability or variability of customer demand for a product or service over a certain period of time. In the aerospace spare parts industry, demand volatility can occur due to changes in aircraft design, maintenance requirements, accidents and incidents, as well as shifts in customer demand and economic conditions. Managing demand volatility in this industry is challenging due to the long-life cycles of primary products, the complexity of managing many types of spare parts, the intermittent demand, and the stochastic nature of demand. Strategies for managing demand volatility in the aerospace spare parts industry include using demand forecasting tools, implementing flexible manufacturing processes, improving supplier collaboration, and optimizing inventory levels. Despite these strategies, the stochastic nature of spare parts and the lack of availability can lead to equipment downtime, customer dissatisfaction, and possible penalty costs for after-sales service providers. Therefore, there is a need for continued research and innovation in this industry to improve inventory management and meet customer demands efficiently.

The proposed framework provides a decision-making process for the aerospace spare parts industry to determine the applicability of AM to isolate volatility in demand. The framework considers three decisive factors, including the suitability of using AM for the surveyed parts, the level of demand fluctuation experienced by each product, and the expected production volume. By estimating the Coefficient of Variation (CV) using Olhager's method, if the CV is above the threshold of 40%, AM can be considered a viable option with different rollout scenarios. The framework helps to choose the right rollout scenario, such as combinational deployment strategies, to employ AM for products with high volatility and CM methods for products with stable demand. Additionally, AM can serve as an effective option when the current production and inventory policy underestimates future demand volume, building on the characteristic of on-demand production. Overall, this framework can aid decision-makers in determining the feasibility of using AM to address volatility in demand and optimize production processes in the aerospace spare parts industry.

The resultant framework indicates that AM can be a viable option for producing spare parts in the aerospace industry when demand volatility is high and production volumes are low. The combinational deployment strategy, which involves using AM for products with high volatility and CM methods for products with stable demand, was found to be an effective option. The significance of these results lies in the potential for AM to reduce inventory costs and improve supply chain efficiency in the aerospace industry. Some limitations, however, are inherent to the conceptual work offered here and should be acknowledged. One of the primary limitations is that this work does not include empirical data to support the proposed concepts. As a result, the framework presented is subject to uncertainty and may not always be generalizable to other contexts. Despite this limitation, this conceptual work can serve as a foundation for further empirical research and can provide valuable insights into theoretical and practical issues in this domain.

**Author Contributions:** Conceptualization, A.A.A., A.A. (Ateyah Alzahrani) and A.A. (Ahmad Algamdi); methodology, A.A.A.; software, A.A.A.; validation, A.A.A., A.A. (Ateyah Alzahrani) and A.A. (Ahmad Algamdi); formal analysis, A.A.A.; investigation, A.A.A.; resources, A.A.A.; data curation, A.A.A.; writing—original draft preparation, A.A.A. and A.A. (Ateyah Alzahrani); writing—review and editing, A.A.A., A.A. (Ateyah Alzahrani) and A.A. (Ahmad Algamdi); visualization, A.A.A.; supervision,

A.A.A.; project administration, A.A.A.; funding acquisition, A.A.A., A.A. (Ateyah Alzahrani) and A.A. (Ahmad Algamdi). All authors have read and agreed to the published version of the manuscript.

**Funding:** This research received no external funding.

**Data Availability Statement:** No new data were created or analyzed in this study. Data sharing is not applicable to this article.

**Conflicts of Interest:** The authors declare no conflict of interest.

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
