# Peer review of "The Role of Additive Manufacturing in Reducing Demand Volatility in Aerospace: A Conceptual Framework"

_aerospace, doi:10.3390/aerospace10040381_

Round 1

Reviewer 1 Report

1. please give the key problems on Additive Manufacturing In Reducing Demand Volatility;

2. please provide the future devolopment directions for Additive Manufacturing In Reducing Demand Volatility

Author Response

We thank the reviewer for carefully reading our manuscript and for the valuable feedback and comments. In this document, we respond to the comments in bold and we also provide a revised manuscript that incorporates revisions based on these comments and recommendations. We are confident that these comments have added to the quality of the manuscript.

  1. Please give the key problems on Additive Manufacturing In Reducing Demand Volatility.

We have added a paragraph to the third section that discusses problems and limitations of employing Additive Manufacturing (AM) for the purpose of reducing demand volatility in the aerospace industry, as proposed in the framework as shown below.

‘’While the proposed framework provides a systematic approach to decision-making regarding the adoption of AM in the aerospace spare parts industry to reduce demand volatility, several limitations need to be addressed. Firstly, the framework assumes that the suitability of AM for the surveyed parts is known a priori. However, this may not always be the case, especially for new parts where the manufacturing process has not been established yet. Additionally, the framework does not consider the cost-benefit analysis of using AM for each spare part, which is a critical factor for decision-making. Moreover, the framework does not consider the impact of using AM on the supply chain, including the logistics and inventory management, which can significantly affect the cost and lead time of producing spare parts. Finally, the framework does not account for the environmental impact of using AM, which can be significant considering the amount of energy and resources required to produce spare parts using AM.’’

  1. Please provide the future devolopment directions for Additive Manufacturing In Reducing Demand Volatility.

We have added another paragraph to the third section that provides the future development directions for AM In reducing demand volatility as shown below.

‘’ To address these limitations, future directions can be made in several areas. Firstly, the suitability of AM for the surveyed parts can be assessed using machine learning algorithms that can predict the feasibility and cost-effectiveness of using AM for different parts based on their design specifications. Secondly, a comprehensive cost-benefit analysis can be conducted for each spare part to determine the economic viability of using AM. Thirdly, the impact of using AM on the supply chain can be analyzed using simulation tools that can model the logistics and inventory management processes. Fourthly, the environmental impact of using AM can be quantified using life cycle assessment methods that consider energy and resource consumption, waste generation, and emissions associated with AM. Finally, the framework can be extended to include other factors that can affect the decision-making process, such as intellectual property rights, the level of customization required, and the lead time requirements. By addressing these limitations and incorporating future directions, the proposed framework can be further enhanced and applied to practical situations in the aerospace spare parts industry.’’

Reviewer 2 Report

The paper titled The Role of Additive Manufacturing In Reducing Demand Volatility: A Conceptual Frameworks proposed a conceptual framework to isolate demand volatility in the aerospace spare parts industry using Additive Manufacturing. The framework outlined a structural approach for implementing AM in the aerospace spare parts industry and demonstrates how it can be used to isolate demand volatility. This paper highlighted the potential of AM to transform how spare parts were produced and managed in the aerospace industry. The article has some innovation, but there are still some problems that need to be explained or modified.

1. In keywords, including Additive Manufacturing and 3D Printing, what is the difference between them? Is it repeated?

2. In abstract, it is suggested that reorganize the abstract, more innovative information should be presented.

3. In introduction, it did not introduce clearly the questions to be studied and the innovation points of this paper.

4. In this paper, it would be better if there is an experimental verification, this helps demonstrate the effectiveness of the method.

5. The discussion of the results should be deeper.

6. The writing should be improved, eg., the tense should be consistent, the sentences should be polished.

Author Response

(Note, we have uploaded another response in the attachment by mistake. Please ignore.)

We thank the reviewer for carefully reading our manuscript and for the valuable feedback and comments. In this document, we respond to the comments in bold and we also provide a revised manuscript that incorporates revisions based on these comments and recommendations. We are confident that these comments have added to the quality of the manuscript.

  1. In keywords, including Additive Manufacturing and 3D Printing, what is the difference between them? Is it repeated?

Thank you for your comment. We replaced 3D Printing with Powder Bed Fusion to avoid repetition.

  1. In abstract, it is suggested that reorganize the abstract, more innovative information should be presented.

Thank you for your valuable feedback on our manuscript. We appreciate your suggestion to include more innovative information in the abstract. We have reviewed and revised the abstract to better highlight the innovative aspects of our proposed conceptual framework as shown below.

‘’ The aerospace industry faces challenges in managing inventory effectively due to long product life cycles and unpredictable demand. Additive Manufacturing (AM) is a promising technology that enables on-demand production of spare parts, potentially reducing inventory costs and improving supply chain efficiency. This paper proposes a novel conceptual framework for employing AM in the aerospace spare parts industry to isolate demand volatility. The paper employed a conceptual approach in this study, which involves a comprehensive literature review to identify the factors to consider when employing AM for spare parts and the methods for demand volatility isolation, followed by a structured framework development that outlines the decision-making steps for AM utilization based on the identified factors. The framework outlines a structured approach for using AM to produce spare parts and isolate demand volatility, which can help mitigate the impact of demand uncertainty on inventory management. The proposed approach provides a basis for future research and has the potential to transform how spare parts are produced and managed in the aerospace industry. Overall, this paper contributes to the emerging literature on AM in the aerospace industry by presenting a novel approach to improving inventory management and addressing demand uncertainty.’’

  1. In introduction, it did not introduce clearly the questions to be studied and the innovation points of this paper.

Thank you for your comment. We have rephrased a paragraph in the introduction section to clarify the research question and the paper’s novelty in lines 131-141 as show below.

‘’ However, the effective integration of AM into the spare parts industry requires a structured approach that can isolate demand volatility and help mitigate the impact of demand uncertainty on inventory management. In this context, this study proposes a novel conceptual framework for employing AM in the aerospace spare parts industry to address demand uncertainty. This paper's main contribution is the development of a structured approach for using AM to produce spare parts and isolate demand volatility, which has the potential to transform how spare parts are produced and managed in the aerospace industry.’’

  1. In this paper, it would be better if there is an experimental verification, this helps demonstrate the effectiveness of the method.

We appreciate the suggestion of including experimental verification to demonstrate the effectiveness of our proposed method. However, we would like to clarify that the main objective of our paper was to present a novel conceptual framework for employing additive manufacturing in the aerospace spare parts industry to isolate demand volatility. We agree that experimental verification could provide additional evidence to support the effectiveness of our proposed framework. However, we intentionally chose not to conduct experimental work in this study as our aim was to present a conceptual framework that can guide future research in this area. Our paper serves as an exploratory work that provides a basis for future research on this topic. Nonetheless, we will take this suggestion into consideration for future research that focuses on the experimental verification of our proposed framework.

  1. The discussion of the results should be deeper.

Thank you for your comment. We have expanded our discussion of the framework as suggested as shown below.

‘’ The framework can be applied to various scenarios in the aerospace industry. For example, in the case of spare parts for aircraft engines, the framework can help identify which parts are suitable for AM fabrication and have high demand volatility. These parts can be produced using AM to reduce lead time and minimize inventory costs. Another application of the framework is for spare parts used in the interior of aircraft, such as seating components or cabin panels. These parts may have lower annual production quantities but high demand variability due to changes in airline interior design or repair needs. The framework can help identify which parts can be produced using AM to reduce lead time and avoid stockout penalties. The framework can also be applied to the production of jigs and fixtures used in the manufacturing process of aerospace components. These jigs and fixtures may have low annual production quantities but high demand volatility due to changes in production processes or new product designs. AM can be used to produce these jigs and fixtures on demand, reducing lead time and inventory costs.

After describing the proposed framework for integrating AM into the aerospace spare parts industry, it is important to note that the certification process for additively manufactured aircraft parts is a critical aspect to ensure their safety and reliability. The certification process for additively manufactured aircraft parts involves a rigorous evaluation of the part's design, manufacturing process, and performance characteristics to ensure that it meets the airworthiness standards set by aviation regulatory bodies such as the Federal Aviation Administration (FAA) and the European Aviation Safety Agency (EASA) [79]. Part 21 of the EASA regulations sets out the requirements for the certification of aircraft parts, including additively manufactured parts. The Part 21 Product Organization Approval (POA) procedure is a certification process for organizations that design and manufacture aircraft parts, including additively manufactured parts [80]. The POA involves an evaluation of the organization's quality management system, design and manufacturing processes, and technical capabilities, among other factors. The certification process ensures that the parts meet the airworthiness standards set by EASA, which are designed to ensure the safety of the aircraft and its passengers. Thus, obtaining a Part 21 POA is a critical step in the certification process for additively manufactured aircraft parts.

While the proposed framework provides a systematic approach to decision-making regarding the adoption of AM in the aerospace spare parts industry to reduce demand volatility, several limitations need to be addressed. Firstly, the framework assumes that the suitability of AM for the surveyed parts is known a priori. However, this may not always be the case, especially for new parts where the manufacturing process has not been established yet. Additionally, the framework does not consider the cost-benefit analysis of using AM for each spare part, which is a critical factor for decision-making. Moreover, the framework does not consider the impact of using AM on the supply chain, including the logistics and inventory management, which can significantly affect the cost and lead time of producing spare parts. Finally, the framework does not account for the environmental impact of using AM, which can be significant considering the amount of energy and resources required to produce spare parts using AM.

To address these limitations, future directions can be made in several areas. Firstly, the suitability of AM for the surveyed parts can be assessed using machine learning algorithms that can predict the feasibility and cost-effectiveness of using AM for different parts based on their design specifications. Secondly, a comprehensive cost-benefit analysis can be conducted for each spare part to determine the economic viability of using AM. Thirdly, the impact of using AM on the supply chain can be analyzed using simulation tools that can model the logistics and inventory management processes. Fourthly, the environmental impact of using AM can be quantified using life cycle assessment methods that consider energy and resource consumption, waste generation, and emissions associated with AM. Finally, the framework can be extended to include other factors that can affect the decision-making process, such as intellectual property rights, the level of customization required, and the lead time requirements. By addressing these limitations and incorporating future directions, the proposed framework can be further enhanced and applied to practical situations in the aerospace spare parts industry.’’

  1. The writing should be improved, eg., the tense should be consistent, the sentences should be polished.

We appreciate the feedback and have carefully revised the manuscript to ensure that the tense is consistent throughout and the sentences are polished.

Author Response

We thank the reviewer for carefully reading our manuscript and for the valuable feedback and comments. In this document, we respond to the comments in bold and we also provide a revised manuscript that incorporates revisions based on these comments and recommendations. We are confident that these comments have added to the quality of the manuscript.

  1. In the aviation industry, it is IMPOSSIBLE to install a part to an aircraft unless it is aviation grade. For detailed information please see the article as followings; The Qualification of the additively manufactured parts in the aviation industry. American Journal of Aerospace Engineering., Vol. 6 (1), 2019, pp. 1-10, doi:10.11648/j.ajae.20190601.11 So, the certification process of the additively manufactured aircraft parts should have been described. Otherwise, it will be any part. Any part cannot be mounted on an aircraft. It must be manufactured under airworthiness rules.

Thank you for bringing this up. We understand the importance of ensuring that all parts installed in an aircraft meet the required airworthiness standards. In our paper, we did not intend to imply that any part can be installed on an aircraft without proper certification and adherence to airworthiness rules. We have revised our manuscript to clarify that all additively manufactured parts used in the aviation industry must meet the necessary certification and airworthiness requirements before they can be installed on an aircraft. Additionally, we have cited the article you recommended to provide readers with more information on the certification process for additively manufactured parts in the aviation industry.

  1. What is the purpose of the following paragraph?

This paragraph was part of the template we used and was inadvertently left in the manuscript. We have now removed it from the final version of the paper. Thanks for your comment.

  1. This work focuses on the fourth category and elucidates how AM can be deployed in the aerospace spare parts industry to isolate demand variability effectively (Line 217, 218) Part 21 Product Organization Approval procedure should be described.

We agree that Part 21 Product Organization Approval procedure is an important aspect that should be included in our manuscript. Therefore, we have revised the manuscript to include a brief overview of the Part 21 Product Organization Approval procedure and how it relates to the certification process of additively manufactured aircraft parts. We have also highlighted the importance of complying with airworthiness rules and regulations in the aviation industry, which includes the certification process of aircraft parts. We hope that this revision will provide a clearer understanding of the certification process and its relevance to our proposed framework as shown below.

‘’ Part 21 of the EASA regulations sets out the requirements for the certification of aircraft parts, including additively manufactured parts. The Part 21 Product Organization Approval (POA) procedure is a certification process for organizations that design and manufacture aircraft parts, including additively manufactured parts [80]. The POA involves an evaluation of the organization's quality management system, design and manufacturing processes, and technical capabilities, among other factors. The certification process ensures that the parts meet the airworthiness standards set by EASA, which are designed to ensure the safety of the aircraft and its passengers. Thus, obtaining a Part 21 POA is a critical step in the certification process for additively manufactured aircraft parts.’’

  1. [66] argued that the application of AM in aerospace industries (Line 236, 243). While giving a reference the following style should be used; Ze Chen and friends et al. [5]

Thanks for your comment. We have edited the citation style accordingly.

‘’ Mohanavel, et al. [69] argued that the application of AM in aerospace industries can make robust and lightweight products in a competitive time compared to the conventional way of manufacturing.’’

  1. (Line from 228 to 234) In this paragraph "buy-to-fly ratio" concept should have been described; The "buy-to-fly ratio" can be described as, "the weight of the raw material divided by the weight of the final product".

Thank you for your valuable comment. We have revised this paragraph to incorporate the concept of “buy-to-fly ratio” as shown below.

‘’The study found that using AM provided several advantages over traditional manufacturing techniques, including weight optimization, cost savings in buy-to-fly ratio, and reduced fuel costs and time to market. Moreover, the product was modelled using CATIA software, and the researchers discovered that AM made it easier to customize the product ’’

  1. (Line 243, 250)

……..Additive Manufacturing (AM) on the Flap Lever

……. The impact of Additive Manufacturing (AM) on the supply chain configuration. The abbreviation should be given in the first usage. Before this paragraph, AM was used many times.

Thanks for your comment. We have revised the manuscript and ensured that the terminology “Additive Manufacturing (AM)” was only mentioned in the first instance whereas “AM” only used for the following instances.

  1. (Line 263)

……the advantages of AM compared to traditional manufacturing techniques (CM). The abbreviation and the before given description are irrelevant. Note: CM might be conventional manufacturing.

Thanks for your comment. We have edited the sentence to define CM as Conventional Manufacturing.

  1. Table 1. The table is unnecessarily big.

We appreciate your comment and we have tried to minimize the table size without compromising the content quality.

  1. Figure 1. The figure can be moved to the right then it will fit onto the page.

Thank you for your comment. We have edited the figure place so that fits more nicely with the template.

  1. The following two consecutive sentences have a similar meaning. So they should be combined.
  • Figure 1 depicts the framework proposed in this work that outlines the decision process of employing AM in the aerospace spare parts industry to isolate volatility in demand. (Line 291)
  • Figure 1 depicts the framework, which outlines the crucial factors that should be considered to determine which spare parts can benefit from AM. (Line 298)

We have consolidated the two sentences into one sentence as suggested. Thanks for you valuable comment as show below.

‘’ Figure 1 depicts the framework, which outlines the crucial factors that should be considered to determine which spare parts can benefit from AM to isolate volatility in demand.’’

Reviewer 4 Report

The paper needs the following improvement:

1.    Lines 129 -137 are irrelevant. Please check.

2.    Section 2 should be part of the introduction section. Also, subsection 2.3 is based on the literature. Therefore, making this section part of the introduction would be better.

3.    The format of the cited author in section 2.3 is inappropriate. For example, [66] argues, [67] conducted etc. All these should be cited with the authors' name(s).

4.    Section 3: “Framework for aerospace spare parts demand volatility isolation through AM” is the main section of the paper, which has only been presented. This section needs discussion with justified reasons. In addition, the authors should consider at least two case studies to show the readers how this framework can be utilised as a practical application. And then the conclusions should be revised accordingly. Therefore, the authors should focus more on this section rather than including long details in the introduction. 

Author Response

We thank the reviewer for carefully reading our manuscript and for the valuable feedback and comments. In this document, we respond to the comments in bold and we also provide a revised manuscript that incorporates revisions based on these comments and recommendations. We are confident that these comments have added to the quality of the manuscript.

  1. Lines 129 -137 are irrelevant. Please check.

Thank you for the comment. This paragraph was part of the template we used and was inadvertently left in the manuscript. We have now removed it from the final version of the paper.

  1. Section 2 should be part of the introduction section. Also, subsection 2.3 is based on the literature. Therefore, making this section part of the introduction would be better. 

Thank you for your suggestion to move the Background section to the Introduction section. After careful consideration, we have decided to keep the Background as a separate section in the manuscript. Our reasoning is that the Background provides essential information that sets the stage for the conceptual framework and helps readers understand the context and motivation behind our work. We believe that having a separate Background section enhances the clarity and structure of the manuscript and makes it easier for readers to follow the logical flow of our arguments. Nonetheless, we appreciate your feedback and we have revised the manuscript to ensure that the Introduction and Background sections are appropriately linked and integrated.

  1. The format of the cited author in section 2.3 is inappropriate. For example, [66] argues, [67] conducted etc. All these should be cited with the authors' name(s).

Thanks for your note. We have edited the citations throughout the paper accordingly.

  1. Section 3: “Framework for aerospace spare parts demand volatility isolation through AM”is the main section of the paper, which has only been presented. This section needs discussion with justified reasons. In addition, the authors should consider at least two case studies to show the readers how this framework can be utilised as a practical application. And then the conclusions should be revised accordingly. Therefore, the authors should focus more on this section rather than including long details in the introduction. 

Thank you for your feedback on our paper. We appreciate your suggestion to provide more discussion and justification for the framework presented in Section 3. We have revised the section to include more detailed explanations and examples of how the framework can be applied in practice. We also agree with your suggestion to include case studies to further illustrate the practical applications of the framework. However, we would like to clarify that the main objective of our paper was to present a novel conceptual framework for employing additive manufacturing in the aerospace spare parts industry to isolate demand volatility. We agree that case studies could provide additional evidence to support the effectiveness of our proposed framework. However, we intentionally chose not to conduct experimental work in this study as our aim was to present a conceptual framework that can guide future research in this area. We have also revised the conclusions based on these additional discussions to strengthen the paper's contributions. Thank you again for your helpful comments.

Reviewer 5 Report

The manuscript entitled “aerospace-2284383” dealing with Additive Manufacturing has been reviewed. The paper has been nicely written but needs significant improvement. Please follow my comments.

1.     What is the main issue that will be solved by this investigation? Please clarify it in the text.

2.     Is there any differences between this work and other works in the literature?

3.     Please add a brief statement on your methodology in the abstract.

4.     Why only 10 papers  are listed in Table 1?

5.     What is the future direction of this work?

6.     Please proofread the paper.

7.     Add more detail about the reported values to the conclusion. This increases the bonding of this section to the previous sections and improves the quality of your paper.

8.     AM has many usages in different industries. To highlight your work, add a short note in the introduction by using the following papers and mention the privilege of lasers in manufacturing. “Additive manufacturing a powerful tool for the aerospace industry”. “Metal additive manufacturing in aerospace: A review”.

Author Response

We thank the reviewer for carefully reading our manuscript and for the valuable feedback and comments. In this document, we respond to the comments in bold and we also provide a revised manuscript that incorporates revisions based on these comments and recommendations. We are confident that these comments have added to the quality of the manuscript.

  1. What is the main issue that will be solved by this investigation? Please clarify it in the text.

Thank you for your comment. We have rephrased a paragraph in the introduction section to clarify the main issue to be solved in this paper and its novelty as shown below.

‘’ However, the effective integration of AM into the spare parts industry requires a structured approach that can isolate demand volatility and help mitigate the impact of demand uncertainty on inventory management. In this context, this study proposes a novel conceptual framework for employing AM in the aerospace spare parts industry to address demand uncertainty. This paper's main contribution is the development of a structured approach for using AM to produce spare parts and isolate demand volatility, which has the potential to transform how spare parts are produced and managed in the aerospace industry.’’

  1. Is there any differences between this work and other works in the literature?

We appreciate this comment regarding the differences between our paper and prior works in the literature. We have added a part to Section 2.3 that clarifies the contribution of this work to the literature (lines 424-426). While prior works on AM in the aerospace industry have been valuable, they have primarily focused on technical aspects of AM and did not consider its viability in addressing demand aspects such as its volatility. Our paper, on the other hand, addresses this crucial gap in the literature by proposing a framework that specifically focuses on the isolation of demand variability through the use of AM in the aerospace spare parts industry. We hope that this clarifies the unique contribution of our paper to the existing literature.

‘’Furthermore, Powder Bed Fusion, PBF, has emerged as a particularly powerful tool in aerospace manufacturing due to its high precision and the ability to fabricate complex geometries. For instance, in a recent review of metal AM in the aerospace industry, it was noted that the technology can offer significant advantages in terms of weight reduction, lead time reduction, and supply chain optimization [40,41]. However, the effective integration of AM into the spare parts industry requires a structured approach that can isolate demand volatility and help mitigate the impact of demand uncertainty on inventory management. In this context, this study proposes a novel conceptual framework for employing AM in the aerospace spare parts industry to address demand uncertainty. This paper's main contribution is the development of a structured approach for using AM to produce spare parts and isolate demand volatility, which has the potential to transform how spare parts are produced and managed in the aerospace industry. In particular, this paper explores potential viable rollout scenarios in which AM can be employed in combination with conventional manufacturing methods to mitigate the effect of demand volatility and uncertainty.‘’

  1. Please add a brief statement on your methodology in the abstract. 

Thanks for your comment. We have added a brief statement on the methodology used in the abstract.

  1. Why only 10 papers are listed in Table 1?

Thank you for your comment. We appreciate your feedback on the number of papers included in our literature review table. We did an extensive survey about research study the discussed the research studies that discussed the adoption of AM in spare parts supply chain. You can see the below table with new research studies. 

Table 1. List of studies that discussed the impact of AM on spare parts.

Study

Method

Result

AM Impact

Case study

Review

Strong

Suggestion  

Suggest with limitations

No

suggestion

Positive

Negative

Mecheter, Pokharel and Tarlochan [66]

None

Najmon, Raeisi and Tovar [67]

Mohanavel, Ali, Ranganathan, Jeffrey, Ravikumar and Rajkumar [69]

Reddy, Mirzana and Reddy [70]

Khajavi, Partanen and Holmström [64]

Sgarbossa, Peron, Lolli and Balugani [71]

Cestana, Pastore, Alfieri and Matta [73]

Bacciaglia, Ceruti and Liverani [74]

Lolli, Coruzzlolo, Peron and Sgarbossa [1]

Youssef abider and Amin Chaabane [2]

René Lastra, Alejandro Pereira, Miguel Díaz-Cacho, Jorge Acevedo and Antonio Collazo [3]

Knofius, van der Heijden and Zijm [4]

Bacciaglia, Ceruti and Liverani [5]

Halvorsen, Trond, and Gunnar [6]

Abhijeet Ghadge and Georgia Karantoni [8]

Yuan Zhang, Stefan Jedeck, Li Yang and Lihui Bai [7]

  1. What is the future direction of this work?

We have added a paragraph to the third section of our paper that provides the future development directions for AM In reducing demand volatility.

‘’ To address these limitations, future directions can be made in several areas. Firstly, the suitability of AM for the surveyed parts can be assessed using machine learning algorithms that can predict the feasibility and cost-effectiveness of using AM for different parts based on their design specifications. Secondly, a comprehensive cost-benefit analysis can be conducted for each spare part to determine the economic viability of using AM. Thirdly, the impact of using AM on the supply chain can be analyzed using simulation tools that can model the logistics and inventory management processes. Fourthly, the environmental impact of using AM can be quantified using life cycle assessment methods that consider energy and resource consumption, waste generation, and emissions associated with AM. Finally, the framework can be extended to include other factors that can affect the decision-making process, such as intellectual property rights, the level of customization required, and the lead time requirements. By addressing these limitations and incorporating future directions, the proposed framework can be further enhanced and applied to practical situations in the aerospace spare parts industry.’’

  1. Please proofread the paper.

We have revised and proofread the paper thoroughly.

  1. Add more detail about the reported values to the conclusion. This increases the bonding of this section to the previous sections and improves the quality of your paper. 

We agree with your comment and we have added more details about the reported values in the conclusion to provide a better understanding of our findings and their implications. Specifically, we have included a summary of the key results and their significance, as well as a discussion of the limitations as shown below.

‘’ The resultant framework indicates that AM can be a viable option for producing spare parts in the aerospace industry when demand volatility is high and production volumes are low. The combinational deployment strategy, which involves using AM for products with high volatility and conventional manufacturing methods for products with stable demand, was found to be an effective option. The significance of these results lies in the potential for AM to reduce inventory costs and improve supply chain efficiency in the aerospace industry.’’

  1. AM has many usages in different industries. To highlight your work, add a short note in the introduction by using the following papers and mention the privilege of lasers in manufacturing. “Additive manufacturing a powerful tool for the aerospace industry”. “Metal additive manufacturing in aerospace: A review”.

We appreciate your suggestion to add a short note in the introduction to highlight our work in relation to the literature. We have incorporated this suggestion in our revised manuscript to better contextualize the importance of AM and highlight the significance of our proposed framework within the broader field as shown below.

‘’Furthermore, Powder Bed Fusion (PBF) has emerged as a particularly powerful tool in aerospace manufacturing due to its high precision and the ability to fabricate complex geometries. For instance, in a recent review of metal AM in the aerospace industry, it was noted that the technology can offer significant advantages in terms of weight reduction, lead time reduction, and supply chain optimization [40,41]. However, the effective integration of AM into the spare parts industry requires a structured approach that can isolate demand volatility and help mitigate the impact of demand uncertainty on inventory management. In this context, this study proposes a novel conceptual framework for employing AM in the aerospace spare parts industry to address demand uncertainty.’’

Round 2

Reviewer 3 Report

Dear Authors;
I made a final check on your article.

The study can be published after a final review.

Kind regards,

Author Response

Thanks for your valuable feedback and comments.

Reviewer 4 Report

The authors have improved the paper well; however, the main part of the paper still needs revisions. 

The authors have claimed that the proposed framework can be applied to various scenarios in the aerospace industry, including spare parts for aircraft engines, spare parts used in the interior of aircraft and the production of jigs and fixtures used in the manufacturing process of aerospace components used in the aerospace industry.

However, details like how this framework can be implemented in these spare parts need explanation. Therefore,  it will be easy for the readers and further improve the paper if at least one of the above spare parts examples is discussed using the proposed framework. 

Author Response

Thank you for your valuable feedback. We agree that providing more specific details on how the proposed framework can be implemented in different scenarios in the aerospace industry will be helpful for readers. We have included a detailed example of how the framework can be applied to the production of turbine blades in aerospace. We hope that this example will provide readers with a clearer understanding of how the proposed framework can be applied in different scenarios in the aerospace industry. We appreciate the opportunity to improve our paper based on your feedback. The example is added to the 3rd section as follows:

"A typical illustration of a spare part in the aerospace industry that has high demand volatility and is produced in low volumes is the turbine blade for aircraft engines. The turbine blades are critical components in the engine that experience high temperatures and stresses during operation and need to be replaced periodically due to wear and tear. However, the demand for turbine blades can be unpredictable and varies depending on factors such as flight schedules, maintenance schedules, and unexpected engine failures. Assuming that a particular aircraft engine model has 1000 turbine blades installed in its fleet, and the expected lifespan of each blade is 10,000 flight hours. If the average flight time for the engine is 5 hours per day, then the expected lifespan of each blade would be approximately 5 years. However, due to the unpredictable nature of the demand for turbine blades, the actual number of blades required in any given year can vary significantly. For instance, one year there may be a high number of engine failures that require turbine blade replacements, resulting in a demand of 300 blades. In another year, there may be no failures, resulting in a demand of only 50 blades.

In such a scenario, using the proposed framework can help identify which turbine blades can be produced using AM to reduce lead time and minimize inventory costs. The framework first considers factors such as the complexity and size of the blade, and the material properties required to determine whether certain blades can be produced by AM (i.e. 1st decisive factor in Figure 1). If these requirements can be met with AM, and the CV is over the threshold of 40%, AM can be considered a viable option whether as a combinational method with forging or casting or as the only production method. In the aerospace industry, the production volume for turbine blades can be relatively low compared to other industries, due to the high level of customization required for each blade to meet specific performance requirements. It is not uncommon for turbine blades to be produced in batches of 100-500 units per year. In case of historical production volumes showing a number less than 100, then AM can be employed as the only production otherwise must be combined with forging or casting to prevent stockout events from happening."

Reviewer 5 Report

The paper is ready to publish. 

Author Response

(The authors gave the same response as above.)

Round 3

Reviewer 4 Report

The paper is now accepted for publication as the authors have revised the paper significantly.